# Molecular polariton electroabsorption

**Chiao-Yu Cheng[1], Nina Krainova[1], Alyssa N. Brigeman[1], Ajay Khanna ®[2], Sapana Shedge[2], Christine Isborn[2], Joel Yuen-Zhou ®[3] & Noel C. Giebink ®[1]** ✉

We investigate electroabsorption (EA) in organic semiconductor microcavities to understand whether strong light-matter coupling non-trivially alters their nonlinear optical [$\chi^{(3)}(\omega,0,0)$] response. Focusing on strongly-absorbing squaraine (SQ) molecules dispersed in a wide-gap host matrix, we find that classical transfer matrix modeling accurately captures the EA response of low concentration SQ microcavities with a vacuum Rabi splitting of $\hbar\Omega\approx200$ meV, but fails for high concentration cavities with $\hbar\Omega\approx420$ meV. Rather than new physics in the ultrastrong coupling regime, however, we attribute the discrepancy at high SQ concentration to a nearly dark H-aggregate state below the SQ exciton transition, which goes undetected in the optical constant dispersion on which the transfer matrix model is based, but nonetheless interacts with and enhances the EA response of the lower polariton mode. These results indicate that strong coupling can be used to manipulate EA (and presumably other optical nonlinearities) from organic microcavities by controlling the energy of polariton modes relative to other states in the system, but it does not alter the intrinsic optical nonlinearity of the organic semiconductor inside the cavity.

Strong coupling (SC) between light and molecular electronic (or vibrational) transitions has recently driven a fascinating debate over the extent to which nominally intrinsic chemical and photophysical properties of molecules can be altered by placing them in an optical microcavity[1–4]. Experimental reports range from modified chemical reaction kinetics[5,6] to changes in charge carrier mobility[7], photo-induced electron transfer[8], energy transfer[9], and intersystem crossing[10,11], all with varying degrees of theoretical justification[12–19]. Optical nonlinearity is another area of interest[20–23], where some experiments have found large enhancements in second harmonic generation[24] and nonlinear absorption/refraction[25,26] that cannot be reconciled from the optical field enhancement in the cavity, while others have concluded the opposite[27,28].

In this context, electroabsorption (EA)[29] presents an interesting opportunity to explore nonlinearity (EA corresponds to the imaginary part of $\chi^{(3)}(\omega,0,0)$)[30,31] in the SC regime because it is carried out at low light intensity (i.e. the condition under which most vacuum field effects are observed), the metal cavity mirrors conveniently dual as electrodes, and the results can be modeled exactly with classical transfer matrix theory[32]. This last point is noteworthy because it

provides an opportunity to test whether SC actually alters $\chi^{(3)}(\omega,0,0)$ in a way that cannot be understood from the underlying material response (i.e. that of the organic film subjected to the same electric field outside of the cavity) viewed through the filter of the cavity. This question is important because it goes to the core of polariton chemistry, namely whether collective SC with many molecules (which is well described by classical electrodynamics based on the dielectric response of the molecular ensemble) can meaningfully alter the electronic structure/properties/interactions of individual molecules (relevant for local processes such as, e.g. chemical reactions)[1–3].

Here, we address this question by examining EA from a strongly absorbing squaraine (SQ) dye dispersed in a wide-gap host matrix when the SQ exciton transition is weakly and strongly coupled to a Fabry-Perot microcavity mode. We find that transfer matrix modeling accurately captures the EA of low concentration SQ cavities with a vacuum Rabi splitting of $\hbar\Omega\approx200$ meV, but fails for high concentration cavities with $\hbar\Omega\approx420$ meV. Rather than new physics in the ultrastrong coupling regime, however, we attribute the discrepancy in the large Rabi splitting case to the presence of a nearly dark H-aggregate state that forms below the SQ exciton transition. This dark state, which goes

[1]Department of Electrical Engineering, The Pennsylvania State University, University Park, PA 16802, USA. [2]Department of Chemistry and Biochemistry, University of California Merced, Merced, CA 95343, USA. [3]Department of Chemistry and Biochemistry, University of California San Diego, La Jolla, CA 92093, USA. ✉e-mail: ncg2@psu.edu

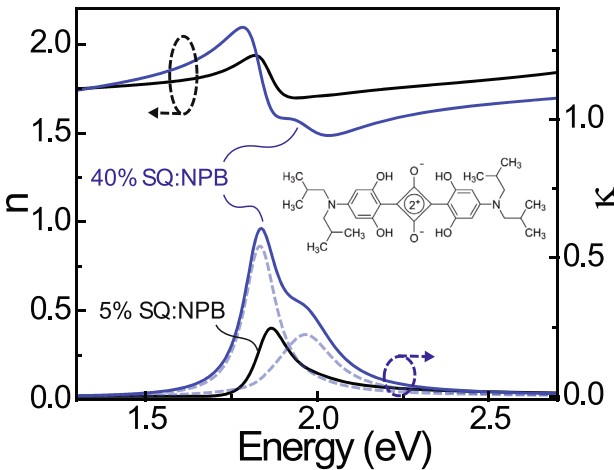

**Fig. 1 | Optical constants of low and high concentration SQ:NPB films.** Complex refractive index dispersion measured for blend films of 5 wt% (black lines) and 40 wt% (blue lines) SQ:NPB via spectroscopic ellipsometry; arrows denote the axis associated with the real ($n$) and imaginary ($\kappa$) parts of the refractive index. The SQ absorption in both cases is modeled using two Lorentz oscillators, which are highlighted for the 40 wt% blend with dashed lines. The SQ molecular structure is shown in the inset.

undetected in the optical constant dispersion on which the transfer matrix model is based (thereby leading to its apparent failure), nevertheless leads to a significant enhancement in the lower polariton EA when the two become resonant, thus emphasizing the impact that seemingly small errors in the input optical constants can have on the interpretation of weak nonlinear signals such as EA. These results indicate that SC can be used to manipulate EA (and presumably other optical nonlinearities) from organic microcavities by engineering the energy of polariton modes relative to other states in the system, but does not alter the intrinsic nonlinear response of the organic semiconductor inside the cavity.

## Results

### Microcavity electroreflectance
Figure 1 shows the optical constant spectra measured for films of SQ co-evaporated with the wide gap host material *N,N*-Di(1-naphthyl)-*N,N*-diphenyl-(1,1-biphenyl)-4,4-diamine (NPB) at concentrations of 5 and 40 wt%. Both blends exhibit a strong transition near 1.85 eV that corresponds to the 0-0 Frenkel exciton transition of SQ. The high energy shoulder that grows more intense in the high concentration film likely derives from a combination of the 0-1 (monomolecular) SQ vibronic and H-aggregates that form in the blend[33–35]. As discussed in the Supplementary Information, the former likely dominates at low concentration, while the latter dominates at high concentration.

The EA response of the low concentration blend is characterized in the weak coupling regime by measuring the electroreflectance of a half-cavity device consisting of a 160 nm-thick layer of 5 wt% SQ:NPB sandwiched between a 120-nm-thick indium-tin-oxide (ITO) anode and a 20-nm-thick Ag cathode. Figure 2a shows the angle-dependent reflectivity spectra of this device, which are well-described by transfer matrix modeling (red dashed lines) based on the optical constant dispersion from Fig. 1. The electroreflectance spectra shown in Fig. 2b are subsequently acquired by depleting the device under reverse bias (see Supplementary Information section S1 for a typical current-voltage characteristic), adding a sinusoidal dither, and synchronously detecting the associated change in reflectivity at the first harmonic of the modulation frequency.

The first derivative lineshape of the resulting electroreflectance spectra qualitatively implies a red-shift of the SQ exciton transition,

which is understood from the Stark shift in transition energy, $\triangle E = -\Delta \boldsymbol{\mu} \cdot \mathbf{F} - (1/2)\mathbf{F} \cdot \Delta \bar{p} \mathbf{F}$, that arises from the difference in static dipole moment ($\Delta \boldsymbol{\mu}$) and polarizability tensor ($\Delta \bar{p}$) between the SQ ground and excited states[29,36]. For a randomly oriented ensemble of SQ molecules, the polarizability term describes a shift in transition energy (e.g. a more polarizable excited state is stabilized by the field, leading to a red-shift), while the static dipole term describes a broadening (molecules where $\Delta \boldsymbol{\mu}$ aligns with the field red-shift while those where $\Delta \boldsymbol{\mu}$ aligns against the field blue-shift). Given the importance of accounting for interference effects as well as the refractive index change that accompanies a shifting exciton transition via the Kramers-Kronig relation[37,38], we dispense with the usual Taylor expansion treatment of EA[29,36,38] and instead calculate the differential reflectivity exactly via the transfer matrix method by adjusting the Lorentz oscillator parameters (energy, amplitude, and broadening) that define the SQ optical constant dispersion in Fig. 1; see Supplementary Information section S2 for details. Using this approach, we find that the electroreflectance spectra in Fig. 2b can be reproduced at all incidence angles in Fig. 2c by red-shifting the 0-0 and 0-1 SQ oscillators (both by 50 μeV) while also strengthening the former (fractional increase in oscillator amplitude, $\Delta f/f = 3 \times 10^{-4}$).

Figure 2d–i present analogous sets of data for two glass/ITO (120 nm)/Al (40 nm)/MoO$_3$ (5 nm)/5 wt% SQ:NPB (X nm)/Ag (20 nm) microcavity samples where the cavity mode at zero in-plane wavevector is negatively ($\Delta = -100$ meV; X = 160 nm) and positively ($\Delta = 170$ meV; X = 140 nm) detuned with respect to the 0-0 SQ exciton transition. The angle-dependent reflectivity spectra in each case (Fig. 2d, g) provide clear evidence of SC, with anti-crossing behavior between the upper (UP) and lower polariton (LP) reflectivity dips that is reproduced by the transfer matrix model. Fitting the dispersion of the UP and LP modes to a standard 2×2 coupled oscillator Hamiltonian[39] yields a vacuum Rabi splitting of $\hbar\Omega \approx 200$ meV for both cavities; see Supplementary Information section S3 for details. Figure 2e, f, h, i demonstrate that the transfer matrix model continues to provide a good description of electroreflectance in the SC regime, with reasonable qualitative agreement obtained between the measured and simulated spectra for both cavities across all angles using the same field-perturbed SQ dielectric function as in the weakly-coupled control sample.

The situation is different for the high concentration SQ:NPB cavities shown in Fig. 3. Beginning with a half-cavity control as in the low concentration case (maintaining the same structure except for a 130 nm-thick, 40 wt% SQ:NPB active layer), we obtain reasonable agreement between the measured (Fig. 3b) and transfer matrix-simulated (Fig. 3c) EA spectra by again red-shifting and strengthening the main SQ exciton transition (by 55 μeV and $\Delta f/f = 10^{-4}$, respectively), though in this case a larger red-shift of the high energy shoulder (by 85 μeV) is required to reproduce the data. Negatively- and positively-detuned 40 wt% SQ:NPB microcavities shown in Fig. 3d, g similarly exhibit clear anti-crossing behavior (with $\hbar\Omega \approx 420$ meV) and are well-described via the transfer matrix model. However, the measured and simulated EA spectra (Figs. 3e, h, f, i, respectively) in this case exhibit some key differences, most notably in the angular dependence of the LP EA amplitude for the negatively-detuned cavity, and in the relative magnitude of the UP EA amplitude for the positively-detuned cavity. We emphasize that these discrepancies are not just a result of using the specific oscillator perturbations from the half-cavity control device; indeed, we are unable to find any reasonable combination of energy, amplitude, or broadening changes for the two SQ oscillators that can reproduce the EA spectra of both cavities at all angles.

### Simplified Hamiltonian model
An alternative to treating the cavity EA in terms of the underlying exciton perturbations (i.e. the transfer matrix approach) is instead to

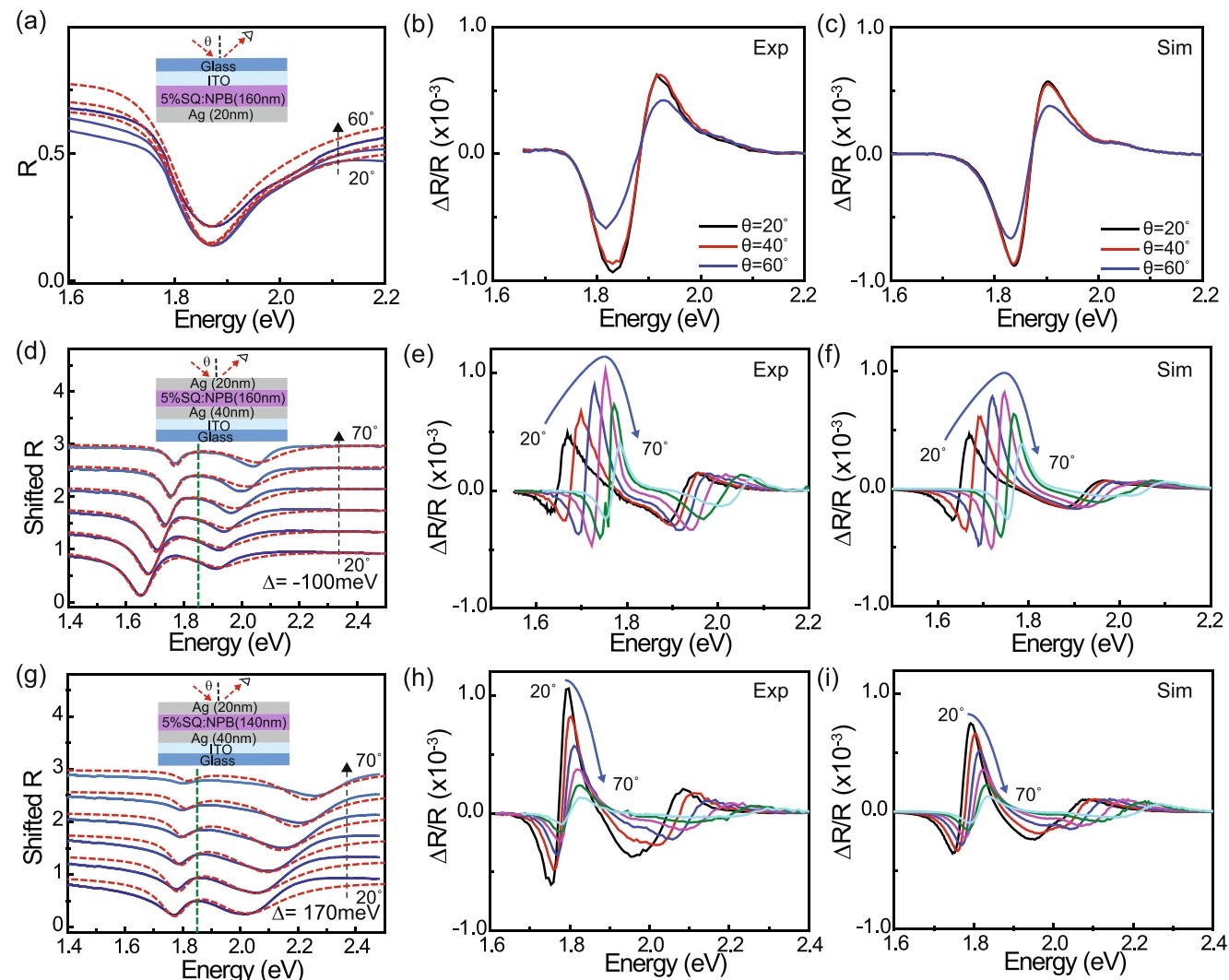

**Fig. 2 | Microcavity electroreflectance at low SQ concentration.** Reflectivity and EA data for devices with a 5 wt% SQ:NPB active layer. **a** Angle-dependent s-polarized reflectivity spectra measured for a half-cavity control device (illustrated in the inset) together with corresponding transfer matrix simulations (red dashed lines) based on the 5 wt% SQ:NPB optical constant dispersion in Fig. 1. **b** Measured and **c** simulated electroreflectance spectra for the control device at different angles. The simulation is carried out via the transfer matrix model by red-shifting and strengthening the Lorentz oscillators used to model the SQ optical constant dispersion as described in the main text. **d** Measured (solid lines) and simulated (dashed lines) s-polarized reflectivity spectra for a negatively-detuned ($\Delta = -100$ meV) 5 wt% SQ:NPB microcavity in the strong coupling regime. The green dashed line indicates the energy of the bare exciton transition. Panels **e** and **f** respectively show measured and simulated electroreflectance spectra for this cavity using the same electric field-perturbed optical constant dispersion as for the half-cavity device in panel c. **g**–**i** Analogous results obtained for a positively-detuned ($\Delta = 170$ meV) microcavity.

characterize it in terms of perturbations to the polariton modes themselves. Given the first derivative lineshape of the LP and UP EA features in Fig. 3e, h, it would appear that the main field-induced perturbation is just a shift in polariton energy (as opposed to, e.g. a change in broadening, which would lead to a second derivative lineshape)[29,36]. In this case, the energy shift ($\triangle E$) can be extracted directly from a Taylor expansion of the reflectivity by scaling the normalized first derivative of the reflectivity, $(1/R)(dR/dE)$, to match the experimental electroreflectance spectra ($\triangle R/R$) for a given LP or UP feature, the proportionality constant being the respective energy shift of that feature; see Supplementary Information section S4 for details. Figure 4 shows that this procedure leads to good agreement with the experimental EA spectra for both cavities, justifying the original assumption that the main effect of the applied field is to shift the polariton energies.

The resulting LP and UP energy shifts are presented for both cavities in Fig. 5c, d, together with those predicted by a simplified

Hamiltonian:

$$H_{SC}(k) = \begin{bmatrix} E_1 & \mu_{12}F & V_1 \\ \mu_{12}F & E_2 & V_2 \\ V_1 & V_2 & E_{ph,k} \end{bmatrix},\qquad(1)$$

that describes coupling between the $S_1$ exciton transition (energy $E_1 = 1.85$ eV) and the cavity photon mode (energy $E_{ph,k}$ at a given in-plane wavevector, $k$) as well as field-induced mixing between $S_1$ and some other excited state at energy $E_2$. This mixing depends on the applied field strength ($F$) and the transition dipole moment between the two excited states ($\mu_{12}$), whereas the coupling of each excited state to the cavity mode depends on its respective transition dipole moment with the ground state via the interaction energies, $V_1$ and $V_2$. Equation (1) is the same coupled oscillator model that is commonly used to describe strong coupling with multiple transitions[40,41], but augmented to include field-induced mixing between them. The associated

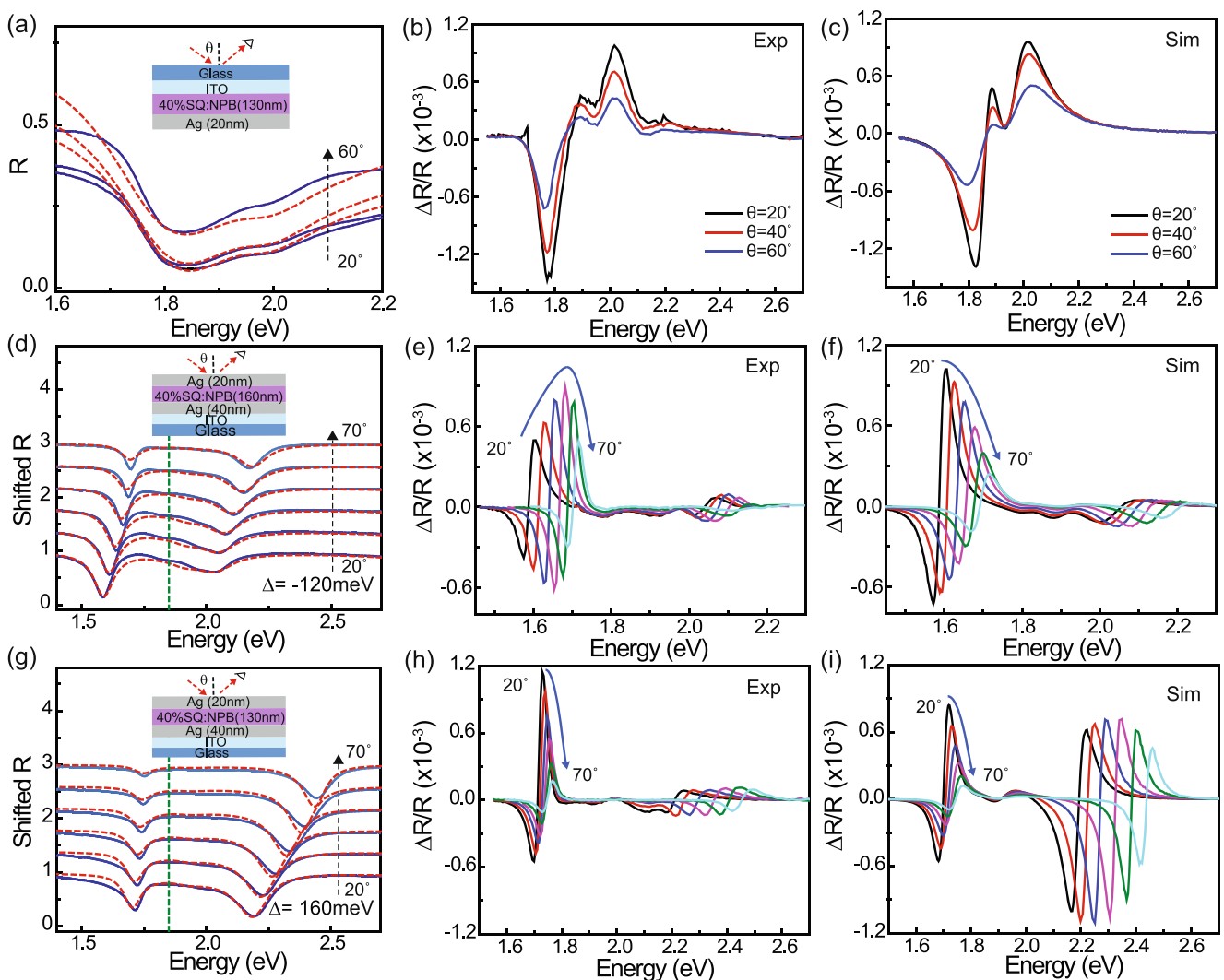

**Fig. 3 | Microcavity electroreflectance at high SQ concentration.** Reflectivity and EA data for devices with a 40 wt% SQ:NPB active layer. **a** Angle-dependent s-polarized reflectivity spectra measured for a half-cavity control device (blue solid lines) together with corresponding transfer matrix simulations (red dashed lines) based on the associated optical constant dispersion in Fig. 1. Panels **b** and **c** show the measured and simulated electroreflectance spectra for this device; the simulation is produced by red-shifting and strengthening the Lorentz oscillators that model the SQ dispersion as described in the main text. **d**–**f** and **g**–**i** show analogous sets of reflectivity, measured, and simulated electroreflectance data for strongly-coupled microcavities with negative ($\Delta = -120$ meV) and positive ($\Delta = 160$ meV) detuning, respectively. The green dashed lines in (**d**, **g**) indicate the energy of the bare exciton transition. In contrast to the experimental EA data in (**e**, **h**), the simulations predict a monotonic decrease in LP EA amplitude for the negatively-detuned cavity (**f**), and a much larger UP-relative-to-LP EA amplitude in the positively-detuned cavity (**i**).

polariton energy shifts are then calculated by diagonalizing Eq. (1) in the presence and absence of the field ($F = 0$). We note that a more precise description of the system would include counterrotating and self-dipole terms accounting for the onset of ultrastrong coupling[42–44]; however, since the electroabsorption effects here are due to near-resonant coupling between states with energy $E_1$ and $E_2$, it seems safe to ignore these terms in the present context.

Assuming initially that $E_2$ corresponds to an upper excited state that lies well above $S_1$ (such as, e.g. the upper excited state of SQ that absorbs at ~3 eV), we can neglect its coupling to the cavity mode ($V_2 = 0$) and fit the polariton dispersion of each cavity in Fig. 5a, b (based on the reflectivity minima in Fig. 3d, g) to determine $V_1 = 0.21$ eV (which yields $\hbar\Omega \approx 2V_1 \approx 0.42$ eV). The remaining unknown ($\mu_{12}$) is then adjusted to match the scale of the experimental energy shifts in Fig. 5c, d, acknowledging that the exact value that results ($\mu_{12} = 20$ D) is not meaningful since it is highly correlated with $E_2$. In any case, although the simulated energy shifts are intuitive insofar as they scale with the matter fraction of each polariton mode, it is clear that this

model fails to capture the much larger LP shift relative to the UP and, in particular, the bump in the LP shift (at ~50°) of the positively-detuned cavity in Fig. 5d.

If, however, we acknowledge that the high energy shoulder of the 40 wt% SQ:NPB blend in Fig. 1 results primarily from H-aggregation based on its concentration dependence (see Supplementary Information section S7)[33–35] and different Stark shift than the main 0-0 transition in the control device, then it implies the existence of a nominally dark, lower H-aggregate state that sits just below the bright $S_1$ exciton. Based on the energy of the upper bright state at ~1.94 eV, we assume that the dark H-aggregate state lies symmetrically below $S_1$ at approximately 1.76 eV. Using $E_2$ to represent this lower aggregate state in Eq. (1), setting $\mu_{12} = 0.25$ D, and allowing for a small, but finite coupling of the aggregate state to the cavity mode ($V_2 = 0.04$ eV; the lower aggregate state is therefore mostly, but not completely dark, as is the case when the molecular transition dipoles are not exactly parallel[45]), we obtain significantly better agreement with the experimental data in Fig. 5e, f. Specifically, the existence of this aggregate state explains the

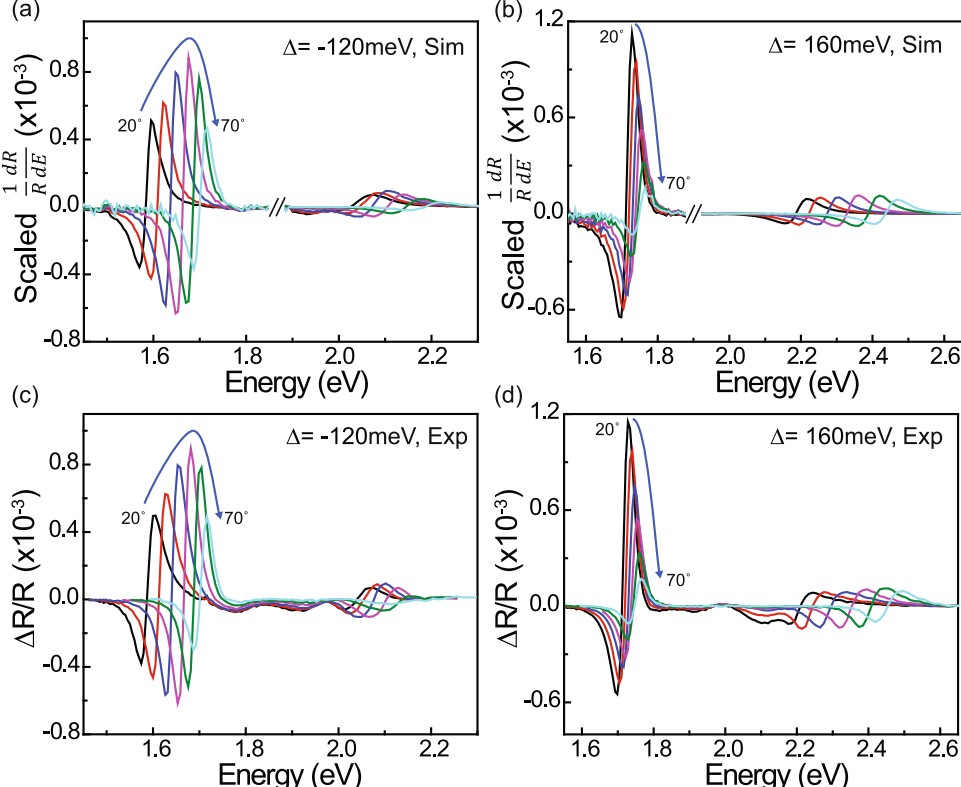

**Fig. 4 | Derivative scaling analysis of polariton electroreflectance.** Scaled derivatives of the experimental reflectivity data, $\frac{1}{R}\left(\frac{dR}{dE}\right)$, for the negatively- (**a**) and positively-detuned (**b**) 40 wt% SQ microcavities. The LP and UP features in **a** and **b** are scaled independently (as indicated by the line breaks at 1.85 eV) to match the amplitude of the corresponding experimental electroreflectance spectra in **c** and **d**; the scaling factors correspond to the field-induced UP and LP energy shifts at each angle as discussed in the text. Note that **c**, **d** are reproduced from Fig. 3e, h for ease of comparison.

more rapid growth of the LP energy shift with increasing angle in the negatively-detuned cavity (compare Fig. 5e vs. 5c), and it predicts the bump in LP energy shift observed for the positively-detuned cavity (Fig. 5f).

Both effects result from the interaction of the LP mode with the aggregate state: whenever the LP approaches the aggregate state (at high angle in the negatively-detuned cavity; see Fig. 5a) or crosses it (at ~50° in the positively-detuned cavity; see Fig. 5b), the LP shift is enhanced. This effect is highlighted in Supplementary Information section S9 by varying the aggregate state energy to change the crossing point, which causes the bump in the LP shift to follow suit, or by reducing the aggregate-cavity coupling strength ($V_2$), which causes the bump to grow sharper. We note that expanding Eq. (1) to include the upper H-aggregate state further improves the agreement with the data (mainly for the UP shift; see Supplementary Information section S8 for details), though it does not change the basic picture established by the two-level model above.

## Discussion

Although Eq. (1) is heavily simplified, the fact that we can explain the peculiarities of the LP shift in both cavities by assuming a single state below $S_1$ with all the characteristics of the lower H-aggregate (i.e. at the expected energy, with low oscillator strength and a physically reasonable excited state transition dipole) strongly supports the aggregate hypothesis. It explains why the transfer matrix treatment fails to describe the EA data from the 40 wt% SQ:NPB cavities (because the nearly dark lower aggregate state goes undetected by ellipsometry and therefore is not included in the optical constant dispersion), but works for the 5 wt% cavities (because the SQ concentration is low enough that H-aggregation is negligible). Thus, the reason that the transfer matrix description fails in the former case is not necessarily because of new

physics in the ultrastrong coupling regime ($\hbar\Omega/E_1 \sim 0.2$ for the 40 wt% cavities), but rather because it is working with incomplete information based on an experimental refractive index dispersion that does not capture the existence of dark states (which may nevertheless gain oscillator strength in an applied field to become relevant in EA). Note, that the term dark here refers to inherently dark molecular transitions outside of the cavity (such as the lower H-aggregate), as opposed to the more common usage in polaritonics associated with linear combinations of molecular transitions that are dark with respect to the cavity.

It is not initially obvious that a transfer matrix description of polariton EA should work to begin with, given that its implicit order of operation (first perturb the exciton, then solve for the polaritons) is opposite that prescribed by perturbation theory (first solve for the polaritons, then perturb them since the exciton-photon interaction is much larger than that of the applied electric field). Nevertheless, at least when all of the states in a system are known and included in an exact Hamiltonian treatment, the two approaches lead to equivalent results (see Supplementary Information section S6), as expected from the fact that the polariton versus exciton+cavity picture just represents a change of basis. In the polariton picture of Fig. 5, the EA response of the LP is amplified when it resonantly interacts with the nearly dark H-aggregate state. In the exciton+cavity picture, the applied field brightens the lower H-aggregate (by mixing it with $S_1$), and this intrinsic material response simply becomes more visible when it is viewed through a mode of the exciton+cavity system (i.e. a polariton).

Regardless of the perspective one chooses, it seems clear that strong coupling offers a tool to manipulate microcavity EA by engineering the energy of polariton modes relative to other states in the system[26] since perturbative effects generally grow whenever the energy difference between perturbatively-coupled states is small[46]. In

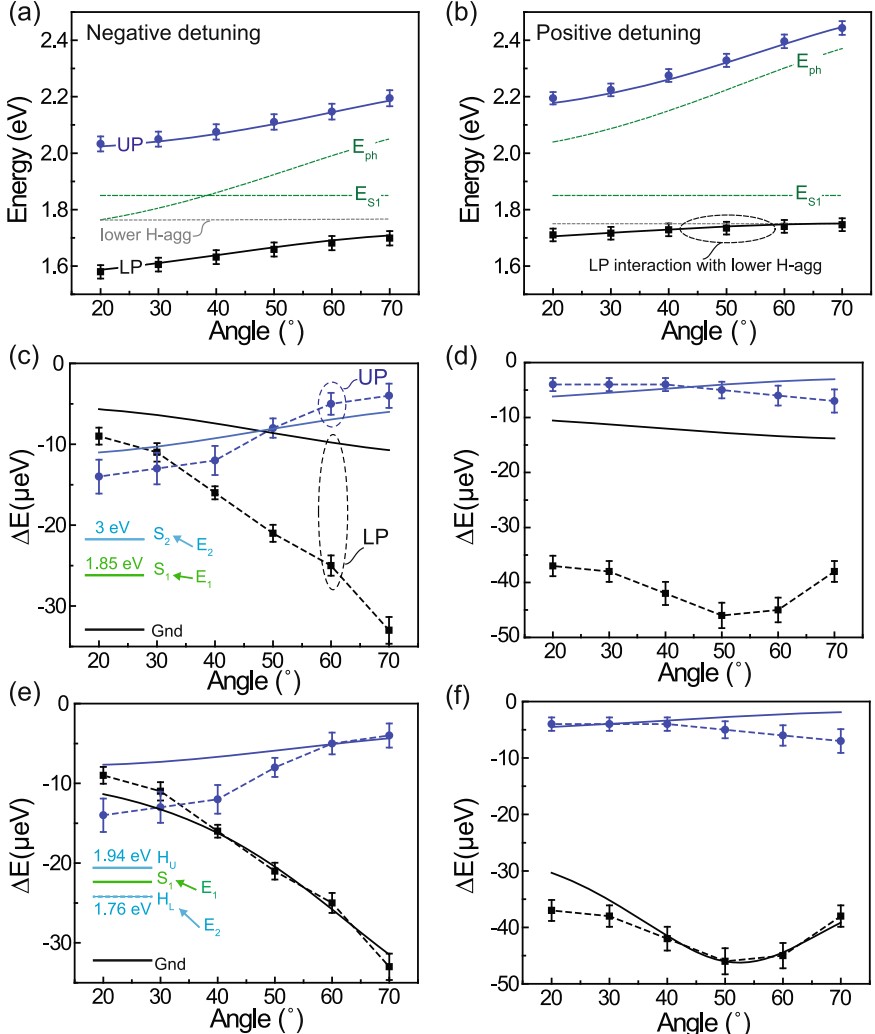

**Fig. 5 | Coupled oscillator model of polariton electroabsorption.** Polariton dispersion relations for the **a** negatively- and **b** positively-detuned 40 wt% SQ:NPB microcavities. The solid markers are determined from fitting the experimental reflectivity minima in each case (error bars reflect the standard error in fitted peak position) and the solid lines are calculated using the coupled oscillator Hamiltonian given in Eq. (1) from the text. The green dashed lines denote the bare cavity mode and SQ exciton energies, while the gray dashed lines reflect the energy of the proposed lower H-aggregate state. Note that there is ostensibly a third, H-aggregate-like branch in the polariton dispersion of each cavity (not shown) that is predicted by Eq. (1) and closely coincides with the lower H-aggregate state (i.e. the gray dashed line); this branch is not observed experimentally because its amplitude is very small (owing to weak coupling of the H-aggregate state to the cavity) and thus is dominated by the much stronger LP feature. **c** Field-induced LP and UP energy shifts derived for the negatively-detuned cavity based on the reflectivity

derivative scaling analysis from Fig. 4a; error bars reflect the uncertainty in the energy shift required to match each experimental EA lineshape. The solid lines show the predicted shifts from Eq. (1) assuming that the second state in the model ($E_2$) corresponds to an upper-excited state of SQ located at 3 eV as sketched in the inset. **d** Corresponding results for the positively-detuned cavity; all of the modeling results assume an applied field strength of $10^5$ V cm$^{-1}$. The same experimental data are shown in **e**, **f**, but in this case, the second state in the model is assumed to correspond to the lower H-aggregate state at 1.76 eV as shown in the inset of **e**. This model explains the growing Stark shift of the LP as it approaches the H-aggregate in the negatively-detuned cavity (**a**, **e**), and the bump in LP shift when it crosses the H-aggregate in the positively-detuned cavity (**b**, **f**). All of the parameters used for the two modeling scenarios (**c**, **d** versus **e**, **f**) are summarized in the Supplementary Information.

addition to the EA enhancement that can occur when a polariton crosses a dark state as in Fig. 5, there is some evidence that SC can alter the usual quadratic field dependence of EA when the cavity couples two nearby bright states that mix in the applied field. This possibility (which follows from Eq. (1)) is explored in Supplementary Information section S10 by strong coupling the doubly degenerate $S_1$ transition of boron subphthalocyanine chloride (SubPc); however, the complexity of this system makes it difficult to reach a firm conclusion. A resolution to this question will likely require a full theory of polariton EA that can predict electroreflectance/transmittance spectra directly from a given input Hamiltonian that includes both light-matter coupling and exciton hopping.

Finally, although our findings here argue against new physics for EA in the SC regime (insofar as it can be understood in terms of the intrinsic material response to the applied field viewed through the filter of the cavity), it is possible that nontrivial SC effects may still manifest for incoherent nonlinearities such as, e.g. saturable or reverse-saturable absorption, where population transfer between different excited states could plausibly be altered by proceeding along a polariton potential energy surface[12]. In this context, it is interesting to speculate about the observations of Ballarini et al.[47], who identified an unusual enhancement in photoluminescence quantum yield for the same strongly coupled SQ:NPB system when the LP minimum is detuned within a narrow energy range below $S_1$ that roughly

corresponds to the LP EA bump in Fig. 5f. The authors explained this phenomenon by postulating the existence of a dark reservoir state that normally drains excitation from $S_1$ (thereby suppressing the luminescence quantum yield of the bare film), but can transfer into the LP mode (and thereby radiate) via a phonon scattering process in the SC regime. Based on our results, it seems likely that this dark reservoir corresponds to the lower H-aggregate, thus offering another example where the intersection of polaritons and dark states can yield interesting effects.

In summary, we have investigated EA from organic microcavities to understand whether SC non-trivially influences their nonlinear optical response. In general, we find that EA in the SC regime is well described via classical transfer matrix modeling based on the intrinsic EA response of the organic film. Apparent discrepancies can arise, however, when the film possesses a nearly dark state (such as the lower H-aggregate in the high concentration SQ blends studied here) that is not accounted for in the transfer matrix model, but which can interact with and enhance the EA of a nearby polariton mode. This enhancement in polariton EA can equivalently be viewed in the exciton+cavity basis as a field-induced brightening of the nearly dark aggregate state (caused by mixing with the bright exciton) that simply becomes more visible when it is viewed through a mode of the exciton+cavity system (i.e. a polariton). These results highlight the fact that interesting and nominally unexpected things can happen in the nonlinear optical response of organic microcavities when a polariton mode interacts with a nearly dark state, but no new physics need be invoked so long as the latter is known and accounted for.

## Methods

2,4-Bis[4-(*N*,*N*-diisobutylamino)-2,6-dihydroxyphenyl] squaraine (SQ), *N*,*N*-Di(1-naphthyl)-*N*,*N*-diphenyl-(1,1-biphenyl)-4,4-diamine (NPB), boron subphthalocyanine chloride (SubPc), and 4,4′-Bis(N-carbazolyl)-1,1′-biphenyl (CBP) are purchased from Sigma-Aldrich and purified by gradient sublimation prior to use. Devices are fabricated on pre-patterned indium-tin-oxide glass that is cleaned with solvents and a 5-minute ultraviolet-ozone treatment before loading into a thermal evaporator with a base pressure of $\sim5 \times 10^{-7}$ Torr. The organic materials are co-evaporated at rates in the range $1–4\,nm\,s^{-1}$ depending on the targeted blend composition, and the metal top contact is deposited through a shadow mask to yield an active device area of $0.1\,cm^2$.

Angle-dependent reflectivity measurements are carried out by mounting samples on a motorized rotation stage and using collimated light from a laser-driven Xe lamp that is filtered through a monochromator and s-polarized with a wire grid polarizer. Electroabsorption is measured using the same setup by depleting the device at $-3\,V$ reverse bias, applying a 0.5 V amplitude sinusoidal dither at 389 Hz, and synchronously detecting the change in reflectivity that results with a lock-in amplifier. Because the EA signal is proportional to the square of the field, $EA \propto F^2 = \left[ F_{DC} + F_{AC} \sin(\omega t) \right]^2$, the signal that we detect at the first harmonic of the modulation frequency is proportional to the product of the AC and DC components, $EA_{1\omega} \propto F_{DC}F_{AC}$. The reflectivity and EA spectra of the half-cavity device are measured through the glass/ITO contact whereas that of the cavity devices is measured through the semitransparent Ag top contact.

## Data availability

The datasets generated during and/or analysed during the current study are available from the corresponding author on request.

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

## Acknowledgements

C-Y. C. and N.C.G. were supported in part by the Charles E. Kaufman Foundation and NSF Grant No. DMR-1654077. J.Y.-Z. was supported by the US Department of Energy, Office of Science, Basic Energy Sciences, CPIMS Program under Early Career Research Program Award DE-SC0019188. S.S., A.K., and C.I. were supported by NSF Grant No. CHE-1955656.

## Author contributions

C.-Y.C. fabricated the microcavity devices, performed the electro-absorption measurements, and carried out the transfer matrix-based analysis of the data; N.K and A.N.B. contributed supporting experiments throughout the project. S.S., A.K., and C.I. carried out the electronic structure calculations and J.Y.-Z. formulated the Hamiltonian-based description of polariton electroabsorption. N.C.G. assisted with the data analysis and C.-Y.C. and N.C.G. wrote the manuscript in consultation with all of the authors.

## Competing interests

The authors declare no competing interests.
