## [Peer Review File · Nature Communications]

Molecular polariton electroabsorptionREVIEWER COMMENTS

Reviewer #1 (Remarks to the Author):

This is an excellent piece of work in a very topical area. I am sure it will be of interest to many readers of Nature Communications.

Cheng et al. report the results of experiments and simulations on a thin film electro-absorption system. This is an interesting choice of system with which to study ensemble strong coupling, the main theme of the report. The authors have clearly taken great care to disentangle strong coupling effects from 'simple' filter effects in their work. This is very important, and is especially welcome as this field enters a more mature and rigorous phase.

I am happy to recommend publication, I make the following observations:

1. I wonder if the authors could suggest any way to independently investigate the presence of dark H-aggregate states in their more concentrated materials?
2. I didn't spend enough time to work out whether the difference in the change in amplitude with increasing angle of incidence for the differential data (see e.g. fig 4a and 4b) was one of the indicators of the failure of the 'simple' model. It might help the reader if a sentence or two could be added to the figure caption.
3. Whilst the focus of the report is on the fact that the authors do not need to invoke new (strong coupling) physics to explain the results they observe, it would nonetheless be interesting if they could offer a view in the conclusion as to whether cavities such as the ones they employ here could become an effective component in any kind of technology.
4. Although the supplementary info is already very extensive, I would like to ask the authors to consider one addition. I would find it useful if they were to plot dispersion diagrams based on their angle dependent reflectivity spectra, e.g. figures 2d, 2g etc..

Reviewer #2 (Remarks to the Author):

In this manuscript, authors demonstrate strong light-matter coupling in organic semiconductor optical cavities can influence electroabsorption. They consider two cases, strong and ultrastrong coupling regimes, and show that EA response of the lower polariton mode gets enhanced in the presence of H-aggregates found in the latter regime. The work performed is of high quality and evidence provided substantiates the claims made. EA in organic polaritons is novel and fitting for Nat Comm. However, I have a few questions that I'd like authors to address/comment.

Authors state that while transfer model can describe the EA response of low 4% SQ concentration (strong coupling) Fabry Perot devices but fails to do so at high 40% SQ concentration (ultrastrong coupling) because H-aggregate is negligible in former. I am interested in knowing whether a SQ doping concentration threshold (and thereby degree of Rabi Splitting and/or V1) can be determined or speculated where H-aggregate is appreciable enough to cause a rapid change in EA response of LP.

Looking at Fig 1, I presume that bandwidth, resonance energy and amplitude of low energy SQ oscillator is changing from low to high SQ concentration. When I think of the crossing of LP and H-aggregate bands, I can't help but dwell on the role of oscillator strength and bandwidth of both on energy transfer between S1, H-aggregate and LP mode. For negatively detuned cavity, do authors observe a dependence of low energy SQ oscillator bandwidth (which could possibly be a function of concentration) on the rapid growth of LP energy shift assuming its bandwidth is largely unchanged? I don't see this aspect discussed in sufficient detail. Can authors comment on this or clarify any misconception I may have here?

Is there any physical relevance to value of $V_2 = 0.04$ eV?

Do authors observe significantly reduced 0-0 PL peak and/or lower quantum yield data for H-aggregate formation at high SQ conc given low energy absorbance is observed around 1.75 eV in Fig 1 that is similar to dark E2 level?

Reviewer #3 (Remarks to the Author):

In this theory-experimental collaborative work by Yuen-Zhou and Giebink, they measured electroabsorption (EA) in organic semiconductor microcavities. To explain the observed polariton dispersion under the very strong coupling regime, they hypothesized that an additional dark H-aggregate state below the exciton S1 state of the system is required to be included in the model Hamiltonian.

With such a model, they have successfully explained the basic feature of the observed dispersion under both negative and positive detuning. The overall message is although the transfer matrix methods failed to explain the experimental results under the ultrastrong coupling regime, it is not due to new physics from the ultrastrong light-matter coupling regime, but rather some other matter states that can complicate the conical pictures of polariton.

The paper also suggests that strong coupling can be used to manipulate EA from organic microcavities by controlling the energy of polariton modes relative to other states in the system.

Overall, I think this is a very nice work of theory and experimental collaboration, which is suitable for publishing in Nature comm. I really like this paper and enjoyed reading it. I do have several concerns that I wish the authors can clarify.

(1) The nature of the S1 exciton states and the dark H-aggregate state.

If I understand it correctly, both of these two states are coming from the collective exciton of molecules (even without the cavity), where the S1 is the bright exciton and the lower energy one is the dark exciton. If so, this needs to be explicitly clarified in the main text.

Further, should V_1 and V_2 be expressed as $g\mu_{01}$ and $g\mu_{02}$, where $|0\rangle$ represents the ground states of matter, g is the collective light matter coupling strength, and μ_{01} and μ_{02} are transition dipoles among the states. If so, then V_1 and V_2 are not independent free parameter, and they need to be careful on fitting the data.

(2) Fitting vs getting the right physics.

Can the authors comment on how many free parameters were used to reproduce the data?

We might all be familiar with the famous comment from Johnny von Neumann (when Fermi told Dyson) “with four parameters I can fit an elephant, and with five I can make him wiggle his trunk”. I do worry that this could all be “a success of fitting”. Can the author think about more predictions that they can make to backup the hypothesis of the presence of dark H-aggregate state?

(3) Building a more microscopic model Hamiltonian.

Related to (2) I think a better approach might be using a more microscopic Hamiltonian (like Holstein–Tavis–Cummings Hamiltonian, eg, in *J. Phys. Chem. Lett.* 2021, 12, 5030–5038). This will not only help to minimize the fitting parameters but also provide more microscopic explanations of the nature of S1 and the dark H-aggregate state. Spano and co-workers might also have the theory built up for J-aggregate coupled to a cavity (eg, in *J. Chem. Phys.* 142, 184707 (2015), which can be used by the authors with some modifications to H-aggregate.

(4) Ultrastrong coupling regime.

Under the collective ultrastrong regime, the Hopfield model should be used instead of the simple Jaynes-Cummings description. I guess this is not included in the model in Eq (1), and should we worry about this?

Reviewer #4 (Remarks to the Author):

This manuscript reports measurements and modeling of the electroabsorption (EA) response of molecular ensembles strongly coupled to an optical cavity. The main importance of the measurement is to test whether the nonlinear response of the molecules is altered qualitatively due to the strong coupling. If there were a non-trivial modification, it would presumably be due to modification of the electronic structure of the molecules. The measurement thus tests the open question of whether electronic (or vibrational) energy structures can be modified in macroscopic molecular ensembles under collective strong coupling to an optical cavity, in a way that leads to measurable changes in properties such as chemical-reaction rates, electronic transport, nonlinear-optical response, etc. This question is controversial, with contradictory experimental results in the literature, and conflicting theoretical arguments about whether such modification is possible in the ensemble. The EA measurements have the advantage of being exactly calculable using a straightforward classical model. The authors have carried out these calculations, with parameters based on control experiments where the molecules are not strongly coupled, and show that there is no change in the EA response due to strong coupling. They also show an enhancement of the EA response when the polariton states are resonant with a dipole-forbidden transition in molecular H-aggregates; this both demonstrates a means to modify response through creating of new resonances, enabled by the energy tuning of strong coupling, and provides an example of an enhancement mechanism whose effects could be mistaken for those of electronic modification. Thus, even though the manuscript reports a "negative" result, it makes an important contribution to a field that is currently of great interest. The experiments and analysis are thoroughly and carefully done, and the conclusions are convincing. I consider the manuscript to be publishable in *Nature Communications* in its present form.

The only minor suggestion that I have concerns the device schematics in the insets of Figs. 2a and 3a. These schematics appear to show the control structures, but no schematics are shown for the full cavity structures, which is somewhat confusing. It would be good to include a similar schematic for the strongly-coupled structures, to go with the corresponding data.

Reviewer Response for Manuscript Entitled:

Molecular polariton electroabsorption

by

Chiao-Yu Cheng, Nina Krainova, Alyssa Brigeman, Sapana Shedge, Ajay Khanna,
Christine Isborn, Joel Yuen-Zhou, and Noel C. Giebink

We thank all of the referees for their feedback and constructive criticism. We have made several revisions to our manuscript according to each point in the referee reports detailed below. Original referee comments are listed in *italics*, our responses are in black, and revisions in the manuscript and supplementary information are highlighted in yellow.

Reviewer #1

I wonder if the authors could suggest any way to independently investigate the presence of dark H-agg states in their more concentrated materials?

We have carried out additional absorption and PL measurements on SQ:NPB films with varying SQ concentration to independently assess the formation of H-aggregate states. Both the absorption and PL spectra exhibit a relative increase and shift of their 0-1 vibronic relative to the 0-0 transition with increasing SQ concentration, which are clear signatures of H-aggregation according to Hestand and Spano in Ref. [45]. The rapid decrease in PL quantum yield that we observe with increasing concentration is also consistent with H-aggregation. We have added these data into a new section in the Supplementary Material as follows:

Figure S7a shows the normalized absorption spectra for a series of 100 nm-thick SQ:NPB films, highlighting an increase in the 0-1 high energy vibronic shoulder relative to the 0-0 transition with increasing SQ concentration. This is accompanied by a relative increase in the 0-1 vibronic photoluminescence intensity (Fig. S7b) as well as a rapid decrease in photoluminescence quantum yield (Fig. S7c). All of these observations are consistent with the signatures of H-aggregate formation described in Ref. [45] and have also been observed for SQ in other host matrices³³. We note that the broadening on the low energy side of the main exciton transition in Fig. S7a can be reproduced by including a weak absorption band located at 1.76 eV that would be consistent with the lower H-aggregate inferred from the polariton EA analysis, though it is not possible to reliably constrain this peak in a fit.

Figure S7. (a) Normalized absorbance spectra measured for 100 nm-thick SQ:NPB films with varying SQ concentration. A 100 nm-thick neat film of NPB (all on glass substrates) is placed in the reference arm of the spectrophotometer to minimize the impact of reflected light when calculating absorbance from transmission. (b) Normalized photoluminescence spectra of the same films using an excitation wavelength of 640 nm. (c) Relative photoluminescence quantum yield (PLQY) determined for these films accounting for the fraction of the excitation beam that is absorbed.

I didn't spend enough time to work out whether the difference in the change in amplitude with increasing angle of incidence for then differential data (see e.g. fig 4a and 4b) was one of the indicators of the failure of the 'simple' model. It might help the reader if a sentence or two could be added to the figure caption.

If the 'simple' model the reviewer is referring to is the transfer matrix model, then the change in amplitude with increasing incidence angle in Fig. 3f,i is indeed one of the indicators of its failure.

In Fig. 4, the angular dependence of the differential data and measured EA data is the same by definition since we have matched the amplitudes to extract the associated

polariton energy shifts. If by ‘simple’ model the reviewer is instead referring to the use of Eqn. (1) assuming involvement of an upper excited state (Fig. 5c,d), then it is the polariton energy shifts that result from Fig. 4 which highlight failure in this case.

Whilst the focus of the report is on the fact that the authors do not need to invoke new (strong coupling) physics to explain the results they observe, it would nonetheless be interesting if they could offer a view in the conclusion as to whether cavities such as the ones they employ here could become an effective component in any kind of technology?

We do speculate a bit on this point in the discussion on page 9 of the manuscript:

“Finally, although our findings here argue against new physics for EA in the SC regime (insofar as it can be understood in terms of the intrinsic material response to the applied field viewed through the filter of the cavity), it is possible that nontrivial SC effects may still manifest for incoherent nonlinearities such as, e.g. saturable or reverse-saturable absorption, where population transfer between different excited states could plausibly be altered by proceeding along a polariton potential energy surface.”

In essence, our conclusion is that while parametric (i.e. instantaneous) nonlinearities do not change in the strong coupling regime, population-based nonlinearities like saturable or reverse-saturable absorption could realistically be altered since they rely on photophysical processes like intersystem crossing and/or internal conversion (e.g. to populate a triplet state) that depend on potential energy surface crossings (which in turn can be influenced through strong coupling via the concept of polariton potential energy surfaces highlighted in Ref. 12). If this proves to be true, it could provide the basis for improving the performance of technologies such as optical limiters and saturable absorbers in a way that cannot be achieved in the weak coupling regime.

Although the supply info is already very extensive, I would like to ask the authors to consider one addition. I would find it useful if they were to plot dispersion diagrams based on their angle dependent reflectivity spectra, e.g. figures 2d, 2g etc..

The dispersion diagrams corresponding to Fig. 2d,g are provided in Fig. S2. The dispersion diagrams corresponding to Fig. 3d,g are provided in Fig. 5a,b.

Reviewer #2

Authors state that while transfer model can describe the EA response of low 4% SQ concentration (strong coupling) Fabry Perot devices but fails to do so at high 40% SQ concentration (ultrastrong coupling) because H-aggregate is negligible in former. I am interested in knowing whether a SQ doping concentration threshold (and thereby degree of Rabi Splitting and/or V1) can be determined or speculated where h-aggregate is appreciable enough to cause a rapid change in EA response of LP.

We have carried out additional experiments on a wider range of SQ:NPB concentrations as described for the first reviewer above. The biggest change in H-aggregation (as inferred from the increase in relative absorption strength of the 0-1 vibronic shoulder) appears to be between 20 and 40 wt% SQ:NPB, though it clearly still continues to increase upon going from 40 wt% SQ:NPB to the neat SQ film.

Looking at Fig 1, I presume that bandwidth, resonance energy and amplitude of low energy SQ oscillator is changing from low to high SQ concentration. When I think of the crossing of LP and H-aggregate bands, I can't help but dwell on the role of oscillator strength and bandwidth of both on energy transfer between S1, H-aggregate and LP mode. For negatively detuned cavity, do authors observe a dependence of low energy SQ oscillator bandwidth (which could possibly be a function of concentration) on the rapid growth of LP energy shift assuming its bandwidth is largely unchanged? I don't see this aspect discussed in sufficient detail. Can authors comment on this or clarify any misconception I may have here?

Energy transfer between, e.g. the LP and H-aggregate state very likely does happen; however, it should not affect the electroabsorption spectra studied here since energy transfer only begins after the initial (instantaneous) absorption event.

To the reviewer's specific question, the rapid growth of the LP energy shift for the negatively detuned cavity (Fig. 5e) is primarily related to the LP approaching the lower H-aggregate state (i.e. it is the beginning of the 'dip' observed in the positively detuned cavity in Fig. 5f) and is not heavily influenced by the broadening of the underlying SQ oscillator (this can be appreciated by using complex energies in Eqn. (1), where the imaginary part reflects the broadening). In general, the main effect of the underlying oscillator bandwidth is to influence the overall amplitude of the EA spectra rather than the changes within a given set of spectra (e.g. as a function of detuning).

To clarify this latter point, we have added the following statement before Fig. S8 in the Supplementary Material:

“Note that the broadening of each transition can also be included in Eqn. (S5) by using complex energies (where the imaginary part of $E_{1,2,3}$ and $E_{ph,k}$ reflects their respective linewidth), though this has little effect on the results shown below and thus is neglected for simplicity.”

2) Is there any physical relevance to value of $V_2 = 0.04$ eV?

Yes, it is proportional to the square root of the lower H-aggregate absorption coefficient since V_2 scales linearly with the corresponding transition dipole moment from the ground state. Thus, $V_2=0.04$ eV together with the bright exciton coupling strength $V_1=0.21$ eV implies that the lower H-aggregate transition is ~ 28 times weaker than the bright exciton transition, which makes it challenging to resolve within the absorption tail of the 40% SQ:NPB blend (though the low energy broadening that occurs in the

concentration-dependent absorption data in Supplementary Fig. S7 is consistent with the existence of such an absorption band as noted for Reviewer #1 above).

3) Do authors observe significantly reduced 0-0 PL peak and/or lower quantum yield data for H-aggregate formation at high SQ conc given low energy absorbance is observed around 1.75 eV in Fig 1 that is similar to dark E2 level?

Yes, we observe both; please see the response to Reviewer #1 above.

Reviewer #3

1) The nature of the S1 exciton states and the dark H-aggregate state. If I understand it correctly, both of these two states are coming from the collective exciton of molecules (even without the cavity), where the S1 is the bright exciton and the lower energy one is the dark exciton. If so, this needs to be explicitly clarified in the main text.

Correct. We have clarified this in the manuscript by adding the following text on page 7:

“If, however, we acknowledge that the high energy shoulder of the 40 wt% SQ:NPB blend in Fig. 1 results primarily from H-aggregation based on its concentration dependence^{33–35} and different Stark shift than the main 0-0 transition in the control device, then it implies the existence of a nominally dark, lower H-aggregate state that sits just below the bright S_1 exciton. Based on the energy of the upper bright state at ~1.94 eV, we assume that the dark H-aggregate state lies symmetrically below S_1 at approximately 1.76 eV.”

Further, should V1 and V2 be expressed as $g\mu_{01}$ and $g\mu_{02}$, where $|0\rangle$ represents the ground states of matter, g is the collective light matter coupling strength, and μ_{01} and μ_{02} are transition dipoles among the states. If so, then V1 and V2 are not independent free parameter, and they need to be careful on fitting the data.?

Correct. Ideally, knowing the relative absorption strengths of transitions 1 and 2 from experiment would fix the ratio of μ_{01} and μ_{02} and better constrain the model. However, because the absorption of the lower H-aggregate state is too weak for us to reliably measure, we are unable to make use of this constraint and are left with the same number of free parameters. Since the light-matter coupling strength is widely written as V_1 , V_2 , etc. in the experimental literature, we chose to adopt that notation here.

2) Fitting vs getting the right physics. Can the authors comment on how many free parameters were used to reproduce the data? We might all familiar with the famous comment from Johnny von Neumann (when Fermi told Dyson) “with four parameters I can fit an elephant, and with five I can make

him wiggle his trunk”. I do worry that this could all be “a success of fitting”. Can author think about more predictions that they can make to backup the hypothesis of the presence of dark H-aggregate state?

This point is well taken. The transfer matrix modeling results in Fig. 2 and Fig. 3 each rely on fits of the respective control film EA spectra, which involve four free parameters (the energy shift and fractional amplitude change of the two oscillators used to model the SQ absorption from Fig. 1). The key point, however, is that there are *no* free parameters going between the control and strongly coupled samples (i.e. the same parameter set is fixed for all of the samples in Fig. 2, and the same for Fig. 3), which makes the central conclusion about whether the material EA response *changes* in the strong coupling regime robust.

In regard to the H-aggregate model in Fig. 5e,f, there are nominally three free parameters: the lower H-aggregate energy E_2 , its coupling to the cavity mode, V_2 , and its transition dipole with the bright exciton, μ_{12} . All of the other parameters in Eqn. (1) are fixed beforehand by the cavity dispersions in Fig. 5a,b, and we further constrain the lower H-aggregate energy E_2 by assuming it lies symmetrically below the bright S_1 exciton based on the position of the upper H-aggregate state inferred from the high energy shoulder in Fig. 1.

Thus, there are only two truly free parameters and they must simultaneously be able to reproduce both the upper and lower polariton shifts of two independent cavities in Fig. 5e,f. There is little ‘wiggle room’ in this case, as evident from Figure S8, which shows that E_2 and V_2 respectively determine the angle and broadening of the ‘dip’ in the LP shift of the positively detuned cavity and thus are well constrained by the data in Fig. 5f. Avoiding a poorly constrained model is precisely the reason that we choose to make use of such a simplified model in Eqn. 1 rather than, e.g. Eqn. S5 that includes the upper aggregate state (and a rapidly multiplying number of free parameters) or the more microscopic extension discussed in the next point below.

3) Building a more microscopic model Hamiltonian.

Related to (2) I think a better approach might be using a more microscopic Hamiltonian (like Holstein–Tavis–Cummings Hamiltonian, eg, in J. Phys. Chem. Lett. 2021, 12, 5030–5038). This will not only help to minimize the fitting parameters but also provide more microscopic explanations of the nature of S_1 and the dark H-aggregate state. Spano and co-worker might also have the theory built up for J-aggregate coupled to a cavity (eg, in J. Chem. Phys. 142, 184707 (2015), which can be used by the authors with some modifications to H-aggregate.

We agree that a Hamiltonian that accounts for vibronic coupling effects would be ideal; however, the present squaraine system is not well suited to implement it from an experimental standpoint because the vibronic structure is not well resolved, which would make it challenging to constrain such a model. A better suited system, which we are currently pursuing, consists of perylene diimide H-aggregates similar to those reported by Brown et al. (*J. Phys. Chem. Lett.*, **5**, 2588 (2014)) in the figure below, which exhibit

well-defined vibronic structure with clear changes in the relative 0-1 to 0-0 peak intensities (which enable extraction of the aggregate coupling strength) that can be used to confidently constrain a full vibronic model. In reference to the development of a more microscopic model to describe a system like this, we have added the suggested references in connection with the following sentence in Section S10 of the Supplementary Material:

“...we emphasize that a full understanding will require a rigorous theory for polariton EA that includes vibronic effects and can be used to fit reflectivity, transmission, and electroreflectance spectra directly from the system Hamiltonian; we are currently

Figure Redacted

48. Qiu, L. *et al.* Molecular Polaritons Generated from Strong Coupling between CdSe Nanoplatelets and a Dielectric Optical Cavity. *J. Phys. Chem. Lett.* **12**, 5030–5038 (2021).
49. Spano, F. C. Optical microcavities enhance the exciton coherence length and eliminate vibronic coupling in J-aggregates. *J. Chem. Phys.* **142**, 184707 (2015).

4) Ultrastrong coupling regime.

Under the collective ultrastrong regime, the Hopfield model should be used instead of the simple Jaynes-Cummings description. I guess this is not included in the model in Eq (1), and should we worry about this?

We thank the reviewer for pointing this out. From the classical optics viewpoint, some terms of the ultrastrong coupling regime are already accounted for in Eq. S2, given the quadratic dependence of the denominator with respect to E_j and E . For the effective Hamiltonians that we have considered, we have ignored such terms because the strongest electroabsorption effects arise from near-resonant static-field-mediated couplings

between states with bare energies E_1 and E_2 . The detuning $|E_1 - E_2|$ is much smaller than the optical gaps, prompting us to safely ignore the ultrastrong coupling effects. In the paragraph right after Eq. 1, we have added the following clarification and references:

"We note that a more precise description of the system would include counterrotating and self-dipole terms accounting for the onset of ultrastrong coupling⁴²⁻⁴⁴; however, since the electroabsorption effects here are due to near-resonant coupling between states with energy E_1 and E_2 , it seems safe to ignore these terms in the present context."

New references:

42. Forn-Díaz, P., Lamata, L., Rico, E., Kono, J. & Solano, E. Ultrastrong coupling regimes of light-matter interaction. *Rev. Mod. Phys.* **91**, 025005 (2019).
43. Kéna-Cohen, S., Maier, S. A. & Bradley, D. D. C. Ultrastrongly Coupled Exciton–Polaritons in Metal-Clad Organic Semiconductor Microcavities. *Adv. Opt. Mater.* **1**, 827–833 (2013).
44. Taylor, M. A. D., Mandal, A., Zhou, W. & Huo, P. Resolution of Gauge Ambiguities in Molecular Cavity Quantum Electrodynamics. *Phys. Rev. Lett.* **125**, 123602 (2020).

Reviewer #4

1) The only minor suggestion that I have concerns the device schematics in the insets of Figs. 2a and 3a. These schematics appear to show the control structures, but no schematics are shown for the full cavity structures, which is somewhat confusing. It would be good to include a similar schematic for the strongly-coupled structures, to go with the corresponding data.

We have added the requested schematics to Figs. 2d,g and 3d,g as shown in the revised figures below:

Figure 2

Figure 3

REVIEWERS' COMMENTS

Reviewer #1 (Remarks to the Author):

I am happy that the authors have responded sufficiently to the comments I raised. My view is that this work now merits publication and discussion, and that Nature Communications is an appropriate journal for that.

Reviewer #2 (Remarks to the Author):

In my view, authors have done a good job in addressing reviewers' comments. The revised manuscript reads well and suitable as is. I recommend publication in Nat Commun.

Reviewer #3 (Remarks to the Author):

I appreciate the effort from the authors to modify the draft and I think all of my original concerns are properly addressed. I thus recommend this paper to be published in Nature Communication.